# Marginal Vertical Fit along the Implant-Abutment Interface: A Microscope Qualitative Analysis

**DOI:** 10.3390/dj4030031

**Published:** 2016-09-06

**Authors:** Nicola Mobilio, Alberto Fasiol, Giulio Franceschetti, Santo Catapano

**Affiliations:** Department of Prosthodontics, Dental School, Dental Clinic, University of Ferrara, Ferrara 44121, Italy; mblncl@unife.it (N.M.); giulio.franceschetti@unife.it (G.F.); cts@unife.it (S.C.)

**Keywords:** abutment, implant, marginal fit, tolerance, optical microscope

## Abstract

The aim of this study was to qualitatively evaluate the marginal vertical fit along two different implant-abutment interfaces: (1) a standard abutment on an implant and (2) a computer-aided-design/computer-aided-machine (CAD/CAM) customized screw-retained crown on an implant. Four groups were compared: three customized screw-retained crowns with three different “tolerance” values (CAD-CAM 0, CAD-CAM +1, CAD-CAM −1) and a standard titanium abutment. Qualitative analysis was carried out using an optical microscope. Results showed a vertical gap significantly different from both CAD-CAM 0 and CAD-CAM −1, while no difference was found between standard abutment and CAD-CAM +1. The set tolerance in producing CAD/CAM screw-retained crowns plays a key role in the final fit.

## 1. Introduction

The implant abutment represents the focus of aesthetic and functional aspects in implant prosthesis, and it affects the long-term prognosis of rehabilitation. The choice of abutment depends on various clinical aspects: implant position, implant angulation, vertical distance between occlusal plane and implant coronal surface, and vertical distance between implant platform and periodontal tissues [1]. Generally speaking, it is possible to choose between: (1) a stock abutment: a standard commercial abutment, not customizable; (2) a grinding abutment: an abutment that can be partially customized by milling but that maintains the standard connection; and (3) a complete customized abutment. The latter may be produced by casting (such as the Universal Clearance Limited Abutment) or may be entirely design and milled (or sintered, melted and so on, depending on materials and techniques) by using the computer-aided-design/computer-aided-machine (CAD/CAM) process.

CAD/CAM abutments are widely used. They undoubtedly present the great advantage of the complete customization of shape, dimension and emergence profile [2,3]. On the contrary, stock abutments present the certainty of coupling between components as designed by the manufacturer [4]. In fact, the precision of the coupling between the implant and abutment may be affected by setting different tolerance values during the CAD/CAM process. In engineering, “tolerance” is defined as the permissible limit of variation in a physical dimension. This variation needs to be acceptable. From a clinical point of view, the tolerance in the coupling between the implant and prosthetic components may affect the fit of the restoration [5]. In implant prosthesis the correct fit is fundamental: an incomplete fit may lead to a marginal horizontal or vertical gap, whereas an excessive fit may become a “misfit” and increases the mechanical overload and, therefore, the risk of failure [6]. Therefore, setting the correct fit during the CAD/CAM process may affect the fit and, potentially, the long-term success of the rehabilitation.

In the literature there are no studies comparing CAD/CAM implant components realized with different tolerance values. The aim of the present study was to qualitatively evaluate the marginal vertical fit along two different implant-abutment interfaces and to compare CAD/CAM screw-retained crowns milled with different “tolerance” values.

## 2. Materials and Methods

Three customized screw-retained crowns were designed (Rhinoceros 5, McNell Europe SL) and milled (RealMeca, Oury Guyè, Dentalmatic, France) from a commercially pure titanium block to fit to an internal hex implant (4 mm diameter, Thommen Medical AG, Switzerland). Each crown was milled with a different “tolerance” value of the inner coupling with the implant connection:
-CAD-CAM 0: tolerance value equal to that declared by manufacturer;-CAD-CAM +1: augmented tolerance (+10 micron);-CAD-CAM −1: reduced tolerance (−10 micron).


A standard titanium grinding abutment (SPI ART, Thommen Medical AG, Switzerland) was used for comparison. The crowns and the standard abutment were connected to implant analogues and screwed at 25 Ncm torque value (Figure 1).

Qualitative analysis was carried out by an optical microscope in polarized radial light with magnification factor of 500% (Leica). A pre-calibrated software for processing high quality digital images (Leica Acquire, Leica Microsystem, Wetzlar, Germany) was used to measure the vertical gap between the implant analogue and the abutment or the crown. Three random measures for each side (buccal, lingual, mesial and distal) were performed. ANOVA was performed to check significance difference (*p* < 0.05) and post-hoc tests were performed to check for inter-groups differences.

## 3. Results

Figure 1 shows the mean values of vertical gaps (micron) for each interface. The standard abutment presented a smaller vertical gap (3.4 ± 1.2) and differs significantly from both CAD-CAM 0 (5.9 ± 1.2) and CAD-CAM −1 (15.1 ± 3.4), while no difference was found with CAD-CAM +1 (4.9 ± 1.3). CAD-CAM −1 showed the largest vertical gap. Figure 2, Figure 3, Figure 4, Figure 5 and Figure 6 show the marginal gap present on one side for each interface.

## 4. Discussion

The abutment-implant interface (such as a screwed crown-implant interface) represents a critical area in implant restoration [3,7].

A misfit between components is recognized as a major concern in implant rehabilitation because it may lead to mechanical problems, first of all relative to the stability of the connection [8,9,10,11,12]. Furthermore, the presence of a microleakage may favor bacterial colonization at the interface, which may cause the inflammation of peri-implant tissues [13,14,15,16]. Obviously, the same problems arise considering the interface between the implant and a screw-retained crown, as analyzed in the present study.

In the present study the vertical gap was measured by randomly choosing three points for each side (buccal, lingual, mesial and distal). This may have reduced the accuracy of the measurement.

Three random points for each side (buccal, lingual, mesial and distal) were randomly chosen for measuring the vertical gap. In the impossibility of measuring the entire perimeter of the interface, 12 measures were assumed to be sufficient to estimate the gap.

The smallest vertical gap (i.e., the best interface) was observed using the stock abutment. Such a result is not unexpected: the quality of the interface as produced by manufacturer is generally recognized. Interestingly, using the same tolerance of the manufacturer, the gap increases. However, the values are below 10 microns, which is generally considered acceptable from a clinical point of view [16]. Increasing the tolerance, the gap does not increase significantly. On the contrary, reducing the tolerance of 10 microns below that declared by the manufacturer dramatically increases the vertical gap. The results showed that CAD/CAM products are not all equivalent: setting different tolerance values leads to crowns (or abutments) that are very different regarding the fitting to the implant.

The present results showed that the largest gap between the implant and crown is found when, during the milling process, tolerance values are set to be lower than those “recommended” by the manufacturer. Many clinicians and dental technicians think that an augmented “grip” between the abutment (or the screwed-retained crown) and the implant means a high precision in the fit at the interface, and they look for such a perceived fit as, in the past, they looked for the die cast being “retained” when being pushed in the metal framework. To obtain such a “fit sensation”, it is possible to reduce the tolerance value during milling. However, that which is interpreted as “fit” is just attrition between the walls of the components. The presented results show that reducing the tolerance of 10 microns lower than the manufacturer values is equal to increasing the attrition and, consequently, the vertical gap between the components. In other words, reducing the tolerance leads to the paradoxical consequence of increasing the misfit. Conversely, it is possible to produce a customized CAD/CAM crown (and abutment) with the same fit as the stock abutment, supposing that the set tolerance was similar to (or greater than) that of the manufacturer.

No other studies have analyzed the vertical gaps between implant components depending on various tolerance values imposed during the CAD/CAM process. Further studies are needed to deeply investigate this important aspect.

## 5. Conclusions

Within the limits of this study, the present results show that the set tolerance in producing CAD/CAM screw-retained crowns plays a key role in the final fit between the crown and implant. Further mechanical tests are needed to deeply evaluate this issue.

## Figures and Tables

**Figure 1 dentistry-04-00031-f001:**
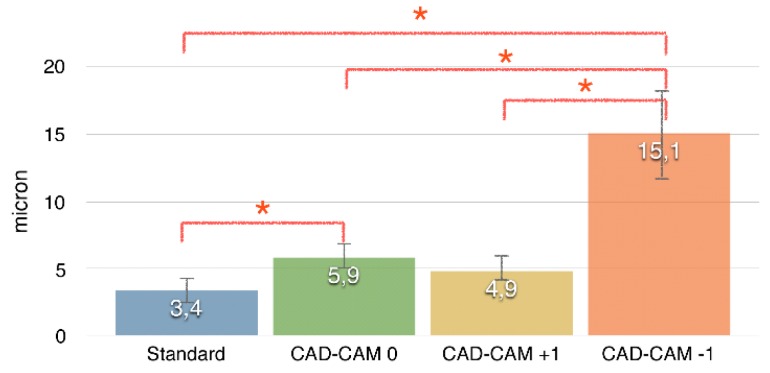
Mean values of vertical gaps (micron) for each interface.

**Figure 2 dentistry-04-00031-f002:**
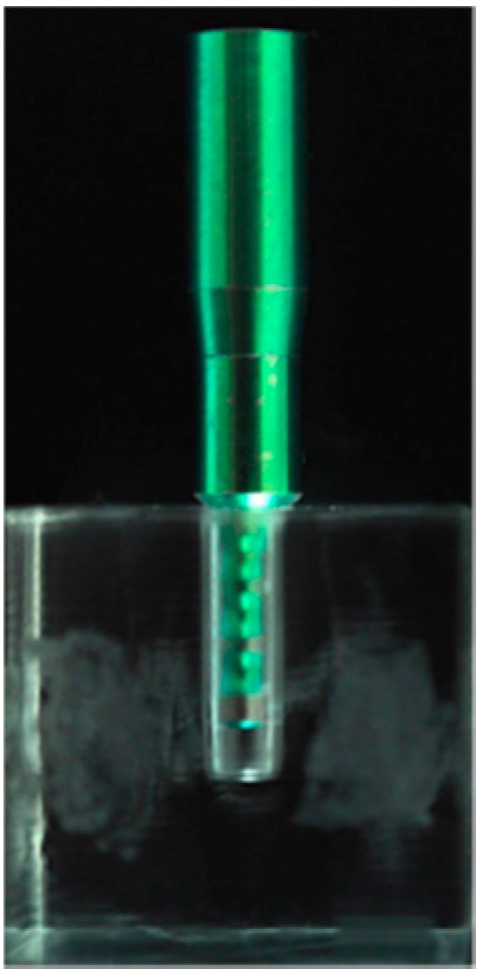
A standard titanium grinding abutment (SPI ART, Thommen Medical AG, Switzerland).

**Figure 3 dentistry-04-00031-f003:**
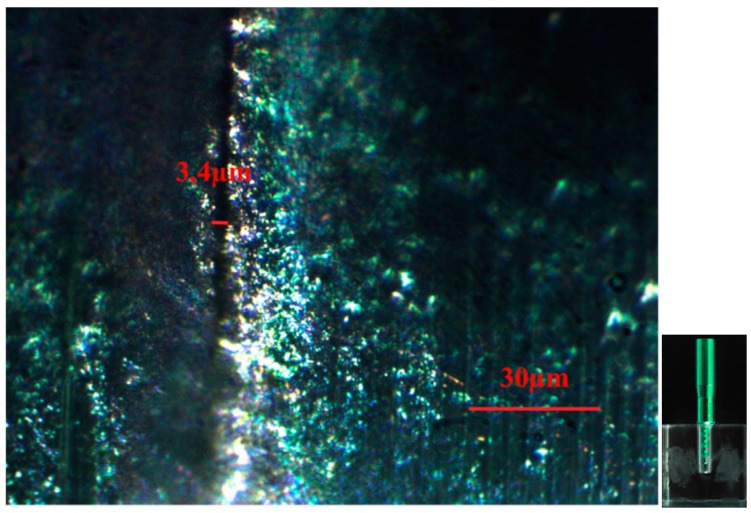
Standard abutment-implant interface shown by an optical microscope in polarized radial light with magnification factor of 500%.

**Figure 4 dentistry-04-00031-f004:**
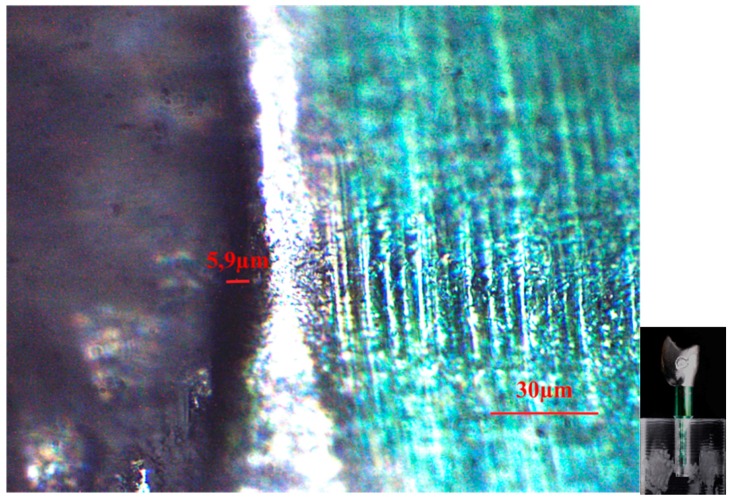
CAD/CAM 0 crown-implant interface shown by an optical microscope in polarized radial light with magnification factor of 500%.

**Figure 5 dentistry-04-00031-f005:**
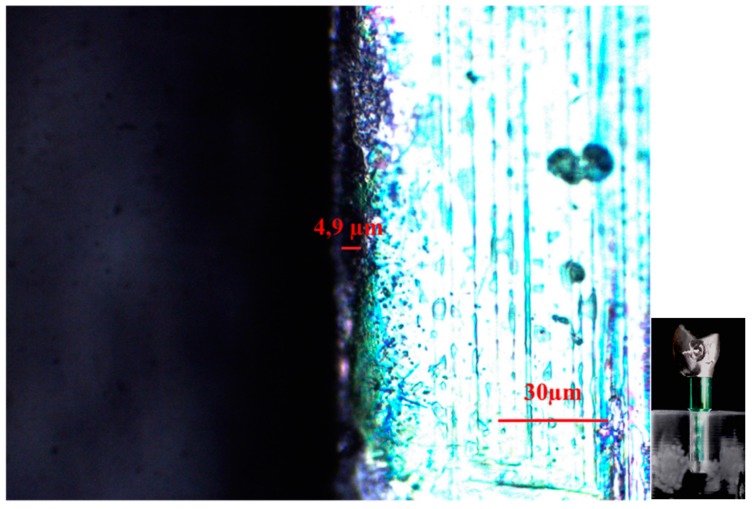
CAD/CAM +1 crown-implant interface shown by an optical microscope in polarized radial light with magnification factor of 500%.

**Figure 6 dentistry-04-00031-f006:**
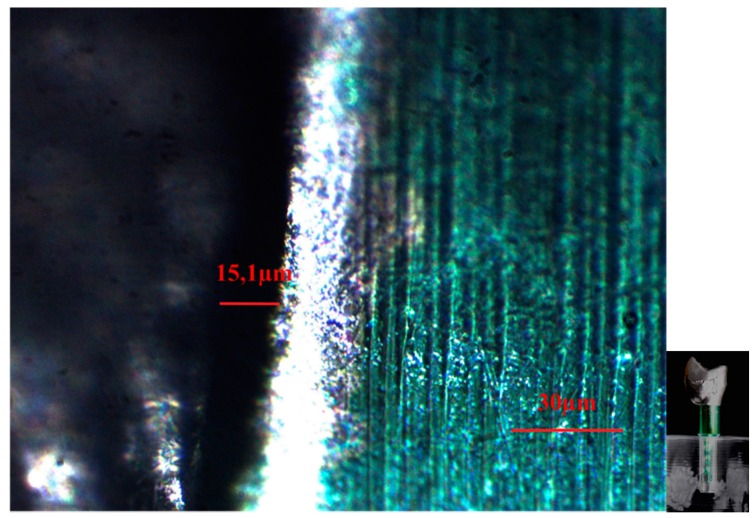
CAD/CAM −1 crown-implant interface shown by an optical microscope in polarized radial light with magnification factor of 500%.

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
