# Peer review of "Marginal Vertical Fit along the Implant-Abutment Interface: A Microscope Qualitative Analysis"

_dentistry, 2016, doi:10.3390/dj4030031_

Round 1

Reviewer 1 Report

Interesting experiment. If possible M&M should be placed as # 2 after the introduction and before the results then the article becomes more straight forward and easy to follow.

Author Response

Materials and Methods section was placed after the introduction.

Reviewer 2 Report

The present study, "Marginal Fit along the Implant-Abutment Interface: An in vitro Qualitative Analysis", covers a topic which is in the focus of Dent. J.. Although the “topic” is of interest in our filed, their appeal to the readership of this journal is too limited to warrant publication. Here are some general remarks that should be addressed by the authors.

TITLE 

- It is difficult to recognize the study from this title. The title should be changed.

ABSTRACT

- The groups investigated are not clear.

- Conclusion is missing in this Abstract. 

INTRODUCTION

- Problem should be completely and accurately explained in the Introduction.- The originality of this study should be clearly stated in the Introduction.

P1L30 They present undoubtedly the great advantage of the complete customization of shape, dimension and emergence profile.

- Please put pertinent reference(s) for this sentence.

RESULTS

- The Result session should be appear after the Materials and Method session.

- The standard deviations of the data are lacking. 

- The results are not presented very well. The authors wrote that they performed an ANOVA, however, the results presented in Figure 1 are suspected to be related to a post hoc comparison.

DISCUSSION

- The Discussion session should be appear at the end.

- The method is not discussed critically. 

- Results are only partially discussed with the results of others.

- Please describe the discussion for the hypotheses provided in this study. 

P4L63 Abutment‐implant interface (like screwed crown‐implant interface) represents a critical area in implant restoration.

- Please put pertinent reference(s) for this sentence.

CONCLUSIONS 

- The conclusions drawn are too general and should be limited to the current study results.

REFERENCES 

- Please check ref#13 in the manuscript.

Author Response

- The title has been changed to better explain the study;
    - in the abstract the groups have been better explained;
    - in the abstract the conclusion was added;
    - The introduction was expanded to better explain the aim of this study;
    - the suggested reference was added;
    - The results session was placed after materials and methods;
    - standard deviations were added to data;
    - The ANOVA and post-hoc comparison were better explained;
    - In the discussion both methods and results were better discussed;
    - the suggested reference was added;
    - The conclusions were changed as suggestedM
    - The ref #13 was checked.

Reviewer 3 Report

Dear Authors;

The manuscript is well written and presented. I think you should mention the rational for choosing three random points on buccal and lingual,... when do the measuring of the fit.

Author Response

The rational for choosing the measuring points was specified in the discussion.

Round 2

Reviewer 2 Report

The authors have not responded appropriately to the comments previously offered by the reviewers. The reviewer kindly asks you to address the reviewers’ criticisms POINT BY POINT as below.

#X INTRODUCTION- Problem should be completely and accurately explained in the Introduction.- The originality of this study should be clearly stated in the Introduction.

(Response)

(Text Change)
